# High Serum Levels of IL-6 Predict Poor Responses in Patients Treated with Pembrolizumab plus Axitinib for Advanced Renal Cell Carcinoma

**DOI:** 10.3390/cancers14235985

**Published:** 2022-12-03

**Authors:** Yun Beom Sang, Hannah Yang, Won Suk Lee, Seung Joon Lee, Seul-Gi Kim, Jaekyung Cheon, Beodeul Kang, Chang Woo Kim, Hong Jae Chon, Chan Kim

**Affiliations:** 1Medical Oncology, Department of Internal Medicine, CHA Bundang Medical Center, CHA University, Seongnam 13496, Republic of Korea; 2Laboratory of Translational Immuno-Oncology, CHA University, Seongnam 13496, Republic of Korea; 3Department of Surgery, Ajou University School of Medicine, Suwon 16500, Republic of Korea

**Keywords:** serum IL-6, predictive biomarker, immunotherapy, renal cell carcinoma, pembrolizumab, axitinib

## Abstract

**Simple Summary:**

Renal cell carcinoma (RCC) is one of the most common types of cancers concerning the kidneys worldwide. Pembrolizumab and axitinib treatment (Pembro/Axi) is amongst the most effective first-line immunotherapies for advanced RCC. However, it remains difficult to predict the effectiveness of Pembro/Axi immunotherapy for the treatment of RCC. Therefore, this prospective study was conducted with the aim of evaluating whether baseline serum interleukin-6 (IL-6) could serve as a predictive biomarker for Pembro/Axi treatment in RCC. Low levels of IL-6 were associated with longer progression-free survival rates, while high IL-6 levels had worse progression-free rates and overall survival. Moreover, high IL-6 levels were related to reduced interferon-γ and tumor necrosis factor-α production. These findings elucidate the clinical and biological implications of IL-6 as a predictive biomarker in RCC patients who received Pembro/Axi as a first-line treatment.

**Abstract:**

Renal cell carcinoma (RCC) is the most common type of kidney malignancy worldwide with Pembrolizumab and axitinib treatment (Pembro/Axi) amongst the most effective first-line immunotherapies for advanced RCC. However, it remains difficult to predict treatment response and early resistance. Therefore, we evaluated whether baseline serum interleukin-6 (IL-6) could be a predictive biomarker. Between November 2019 and December 2021, 58 patients with advanced RCC were enrolled, administered first-line Pembro/Axi, and baseline blood samples were analyzed using flow cytometry. The mean baseline serum IL-6 concentration was 8.6 pg/mL in responders and 84.1 pg/mL in patients with progressive disease. The IL-6 cut-off value was set at 6.5 pg/mL using time-dependent receiver operating characteristic curves, with 37.9% of patients having high baseline serum IL-6 levels and 62.1% having low levels. Objective response rates were 58.3% and 36.4% in low and high IL-6 groups, respectively. Overall survival and progression-free survival were longer in patients with low IL-6 levels than in those with high levels. High IL-6 levels were related to reduced interferon-γ and tumor necrosis factor-α production from CD8+ T cells. Overall, high baseline serum IL-6 levels were associated with worse survival outcomes and reduced T-cell responses in Pembro/Axi-treated advanced RCC patients.

## 1. Introduction

Renal cell carcinoma (RCC) is the most common type of kidney malignancy worldwide [1]. Approximately 30% of patients initially diagnosed with RCC have metastatic disease, and one-third of those subjected to curative treatment, such as surgery, experience recurrence and metastasis [2,3]. Systemic therapies for RCC have significantly advanced in recent years [3]. In a pivotal KEYNOTE-426 trial, pembrolizumab and axitinib (Pembro/Axi), a first-line treatment for RCC patients, showed superior efficacy over sunitinib, regardless of the international metastatic RCC database consortium (IMDC) risk group [4]. Long-term follow-up (median follow-up: over 30 months) data for KEYNOTE-426, showed that the objective response rate for Pembro/Axi was 60.0%, and 86.0% of patients had reduced tumor burden. However, 10.9% of patients treated with Pembro/Axi showed disease progression in their first response evaluation. Moreover, 44.1% of patients had disease progression within one year of treatment initiation [5,6].

There is an unmet need for reliable and clinically feasible biomarkers that can predict treatment response and early resistance of RCC to Pembro/Axi immunotherapy. Although programmed death-ligand 1 (PD-L1) expression in tumors is used to predict the clinical benefits of antiprogrammed cell death protein-1 (PD-1) or PD-L1 immunotherapy in various other malignancies [6,7], it has no role in patients with advanced RCC who are treated with first-line immunotherapy. Moreover, other tissue-based biomarkers, such as microsatellite instability-high/deficient mismatch repair and high tumor mutational burden cannot predict the efficacy of immunotherapy in RCC [8].

Interleukin-6 (IL-6) is a proinflammatory cytokine involved in chronic inflammation [9] and carcinogenesis [10,11]. It is a poor prognostic factor for various cancer types, including melanoma [12,13,14] and RCC [15,16,17]. High serum IL-6 has been demonstrated as a predictor of worse progression-free survival (PFS) benefits in patients with metastatic RCC treated with pazopanib [18]. IL-6, which regulates the function of immune cells, may attenuate antitumor immunity [19]. Myeloid-derived suppressor cells (MDSCs), activated by IL-6, can inhibit T cell responses and promote cancer cell persistence through the JAK/STAT3 signaling pathway [20]. In nonsmall cell lung cancer, IL-6 upregulated PD-L1 expression through the JAK/STAT3 signaling pathway in vitro and increased the recruitment of MDSCs, M2 macrophages, and regulatory T cells (Tregs) while decreasing the infiltration of CD8+ T cells in tumor samples [21]. In bladder cancer, IL-6 enhanced the immune suppression activity of MDSCs via activation of the mitogen-activated protein kinase (MAPK) signaling pathway [22]. A recent in vivo study using an acute lymphoblastic leukemia mouse model showed that IL-6 could suppress chemotherapy-induced anticancer immunity by tumor microenvironment (TME) [23]. However, the immunological effects of IL-6 during immunotherapy remain to be determined.

Here, we evaluated the value of IL-6 in predicting the clinical benefits of first-line Pembro/Axi immunotherapy in patients with advanced RCC.

## 2. Materials and Methods

### 2.1. Patients and Treatments

Between November 2019 and December 2021, 58 patients with advanced or metastatic RCC, who were treated with first line Pembro/Axi, were enrolled (Appendix A). The eligibility criteria included age ≥19 years, recurrent or metastatic clear cell or nonclear cell RCC confirmed by histocytologic diagnosis, no previous systemic treatment, and an Eastern Cooperative Oncology Group (ECOG) performance status of 0–2. All patients received intravenous pembrolizumab at 200 mg every 3 weeks and oral axitinib at 5 mg twice daily. Dose reductions or interruptions were determined according to the KEYNOTE-426 clinical trial protocol [4]. Pembro/Axi treatment was continued until disease progression, intolerable toxicity, or consent withdrawal. Response evaluation was performed every 6 or 9 weeks using computed tomography, magnetic resonance imaging, or bone scan according to Response Evaluation Criteria in Solid Tumors (RECIST) version 1.1. The study was approved by the institutional review board (CHA Bundang Medical Center, 201711054). Written informed consent was obtained from all the patients.

### 2.2. Sample Collection and Measurement of Serum IL-6

Intravenous blood sampling was performed immediately before the first administration of Pembro/Axi. All serum samples were collected in the morning after the overnight fasting. To obtain serum, blood was centrifuged at 1000× *g* for 5 min and stored at −80 °C in a deep freezer. The serum IL-6 was quantified using a cytometric bead array (560484, BD Biosciences, Franklin Lakes, NJ, USA), according to the manufacturer’s instructions [24]. Briefly, the capture beads for human IL-6 were incubated with the serum samples and detection reagent for 3 h at room temperature. After several washing steps, IL-6-bead complexes were measured using a flow cytometer (Beckman Coulter, Brea, CA, USA) and the data were analyzed using FlowJo software version 10.8.1 (Tree Star Inc., San Francisco, CA, USA).

### 2.3. Cytokine Secretion Assays and Flow Cytometry

Peripheral blood mononuclear cells (PBMCs) were obtained by density gradient centrifugation using Ficoll-Paque PLUS (GE Healthcare). To assess cytokine production, PBMCs were activated with a plate bound anti-CD3 antibody (1 µg/mL). After 4 h, brefeldin A (eBioscience) (San Diego, CA, USA) and monensin (eBioscience) were added to stop cytokine secretion, and stimulated cells were harvested after 20 h. Before antibody staining, the cells were stained with Fixable Viability Dye eFluorTM 780 (eBioscience) on ice for 30 min to exclude dead cells, followed by treatment with a human Fc receptor-binding inhibitor (eBioscience) for 15 min at room temperature. Surface proteins were stained with antihuman CD3 (clone SK7, eBioscience), antihuman CD4 (clone RPA-T4, BioLegend) (San Diego, CA, USA), and antihuman CD8 (clone RPA-T8, eBioscience) fluorochrome-conjugated antibodies on ice for 30 min. For intracellular staining, the cells were fixed and permeabilized with a FoxP3 staining buffer kit (Thermo Fisher Scientific) (Waltham, MA, USA) and stained with antihuman interferon-γ antibody (clone B27, BD Bioscience) (San Jose, CA, USA) and antihuman tumor necrosis factor (TNF)-α antibody (clone Mab11, BD Bioscience) fluorochrome-conjugated antibodies. Flow cytometry was performed using a CytoFLEX flow cytometer (Beckman Coulter) (Brea, CA, USA), and the resulting data were analyzed using FlowJo software version 10.8.1 (Tree Star Inc.) (San Francisco, CA, USA).

### 2.4. Analysis of a Public Database

Intratumoral immune cell profiles of clear cell RCC in The Cancer Genome Atlas (TCGA) were analyzed using TIMER version 2.0, a comprehensive resource for the analysis of immune cells across various cancer types. http://timer.cistrome.org (accessed on 10 May 2022) [25].

### 2.5. Statistical Analysis

Comparisons between groups were performed using the chi-square test and the independent sample *t*-test for categorical and continuous data, respectively. Kaplan–Meier analysis was used to estimate survival plots, and the log-rank test was used for comparisons. A Cox proportional hazards model was used to predict PFS and overall survival (OS) after adjusting for other variables. Time-dependent receiver operating characteristic (ROC) analysis was performed to determine the IL-6 cut-off value at the maximum sensitivity and specificity using R 4.2.0 (https://www.r-project.org/) (accessed on 29 April 2022). For all statistical analyses, a *p*-value < 0.05 was considered statistically significant, and calculations were performed using SPSS^®^ Statistics software (Release 18.0).

## 3. Results

### 3.1. Potential Effects of IL-6 on Tumor Immunity

To explore the comprehensive immunological effect of IL-6 on the tumor microenvironment, we analyzed IL-6 levels in immune and stromal cells in clear cell RCC datasets (*n* = 533) from TCGA. Anticancer immune cells, such as CD8+ T cells, CD4+ T cells, and dendritic cells, were not correlated with IL-6 levels except for a weak negative correlation between IL-6 and natural killer (NK) cells (Figure 1a). Intriguingly, IL-6 levels were positively correlated with immunosuppressive cells such as Tregs, MDSCs, and cancer-associated fibroblasts (Figure 1b). Therefore, highly elevated IL-6 levels may be associated with increased intratumoral immunosuppression in patients with kidney cancer. Thus, we investigated whether there were differences in treatment outcomes depending on IL-6 levels in patients with kidney cancer who were administered first line Pembro/Axi.

### 3.2. Patient Demography and Disease Outcome

Fifty-nine patients were enrolled and treated with a combination of pembrolizumab and axitinib. Among them, 58 had baseline blood samples suitable for IL-6 measurement. Although there was no withdrawal of informed consent, three patients discontinued Pembro/Axi treatment due to immune-related adverse events. Baseline demographics are shown in Table 1. The median age of the patient cohort was 60 years (range: 39–83 years) and the male-to-female ratio was 2.6:1. Clear cell histology was observed in 41 (70.7%) patients. The IMDC risk group was favorable for 19.0%, intermediate for 50.0%, and poor for 31.0% of patients. The most common site of metastasis was the lung (81.0%), followed by bone (36.2%), lymph nodes (29.3%), and liver (10.3%). Approximately half of the patients (51.7%) had undergone a previous nephrectomy. All patients were followed up until 20 February 2022. The median follow-up duration was 12.1 months (IQR: 8.0–17.4). The median PFS and OS of all patients were 12.4 (95% confidence interval [CI]: 8.2–16.6) and 21.9 (95% CI: 18.5–25.3) months, respectively.

### 3.3. Baseline Serum IL-6 Levels and Clinical Response to Pembrolizumab/Axitinib

The average baseline concentration of plasma IL-6 was 8.6 pg/mL in patients with objective responses (complete response [CR]/partial response [PR]; *n* = 29), 9.9 pg/mL in patients with stable disease (SD, *n* = 21), and 84.1 pg/mL in patients with progressive disease (PD, *n* = 8; Figure 2a). Next, we determined the cut-off value for serum IL-6 to define patients with high and low levels. Using time dependent ROC curve analysis, a cut-off value of 6.5 pg/mL had 71.3% sensitivity and 87.5% specificity for a predictive 6-month OS rate in patients treated with Pembro/Axi. The area under the curve was 88.5% (Figure 2b). At this cut-off point, 37.9% (*n* = 22) and 62.1% (*n* = 36) of the patients had high and low levels of serum IL-6, respectively, at baseline. After Pembro/Axi treatment, patients in the high IL-6 group showed a remarkably higher rate of PD than those in the low IL-6 group; 31.8 and 2.8%, respectively. The objective response rates were 36.4 and 58.3% in the high and low IL-6 groups, respectively (Figure 2c).

Patients with high IL-6 levels at baseline had more IMDC poor-risk diseases and more frequent lymph node and bone metastases than those with low IL-6 levels (Table 2). Otherwise, there were no significant differences in baseline clinical characteristics between the two groups.

### 3.4. Baseline Serum IL-6 Levels and Survival Outcome with Pembro/Axi Therapy

Survival outcomes were compared for different serum IL-6 levels. Patients with high IL-6 levels had significantly shorter PFS than those with low IL-6 levels (hazard ratio (HR): 3.51, 95% CI: 1.54–7.98, *p* = 0.003; Figure 3a). Although most (72.2%) patients with low IL-6 levels remained progression free at 12 months, only 40.9% with high IL-6 levels were progression free at 12 months, indicating a predictive role of baseline IL-6 levels for Pembro/Axi therapy. Moreover, patients in the IL-6 high group showed worse OS than those in the IL-6 low group (HR: 7.18, 95% CI: 2.26–22.82, *p* = 0.001; Figure 3b).

Because clinicopathological variables, such as IMDC risk group or pathologic type, could be confounding factors for survival outcomes, we conducted multivariate analyses using the Cox proportional hazard model (Figure 3c,d). A high level of IL-6 at baseline was an independent predictive factor for PFS and OS (HR: 2.454, 95% CI: 1.035–5.818, *p* = 0.041, and HR: 5.511, 95% CI: 1.656–18.342, *p* = 0.005, respectively), regardless of a previous history of nephrectomy, a number of metastatic sites, pathologic type, ECOG performance status, and IMDC risk group.

### 3.5. High Serum IL-6 Levels Are Associated with Reduced T-Cell Responses

To evaluate the effects of serum IL-6 on T-cell function, we compared cytokine production between the IL-6 high and low groups at baseline. The IL-6 high group showed a 43.6% decrease in interferon-γ and TNF-α secretion compared to the IL-6 low group (Figure 4). Consequently, high IL-6 concentrations at baseline were associated with reduced CD8+ T cell activation. The average proportions of activated CD8+ T cells were 4.1% and 7.3% in the IL-6 high and low groups, respectively (*p* = 0.009).

## 4. Discussion

In this study, we demonstrated that high baseline serum IL-6 levels are associated with worse clinical outcomes in patients with advanced RCC treated with first line Pembro/Axi. Patients with high baseline serum IL-6 levels showed a lower response rate, a higher PD rate, and inferior survival outcomes for both OS and PFS than those with low baseline IL-6 levels. Notably, a high serum IL-6 level at baseline was an independent predictor of worse clinical outcomes following Pembro/Axi immunotherapy, even after adjusting for potential confounding factors. Moreover, CD8+ T cells from patients with high IL-6 levels produced fewer effector cytokines, IFN-γ and TNF-α, indicating that they were less functional than those from patients with low IL-6 levels.

IL-6 is produced by a variety of cells including T cells, B cells, macrophages, and tumor cells [26]. It is involved in inflammation and immune regulation, cell survival, and growth through the JAK/STAT3 signaling pathway, and regulates the antitumor T cell and innate immune responses by promoting the generation of these immunosuppressive cells [27,28]. Consistent with these preclinical findings, analysis of RCC patient datasets from TCGA revealed a positive association between IL-6 levels and immunosuppressive cells, such as Tregs, MDSCs, and cancer-associated fibroblasts. Notably, NK cells were negatively associated with IL-6 expression, consistent with a previous finding that IL-6 secreted from esophageal squamous carcinoma cells decreased NK-cell activity through the STAT3 signaling pathway in vitro [29].

To the best of our knowledge, this is the first study to clarify the clinical and biological implications of IL-6 as a predictive biomarker in RCC patients receiving Pembro/Axi as a first line treatment. The predictive value of IL-6, when using sunitinib as a first line treatment for metastatic RCC patients, has been evaluated; however, these studies enrolled few patients, only those with clear cell types or with poor IMDC risk [30,31]. In contrast, we enrolled a relatively large number of patients treated with Pembro/Axi, including all histological subtypes and IMDC risks, and elucidated the immunological differences in CD8+ T cells between patients with high and low IL-6 levels. CD8+ T cells from patients with high IL-6 levels were less functional, possibly due to the reduced clinical outcomes of immunotherapy.

However, our study has several limitations. Because the study was conducted at a single center, external validation in a multicenter cohort is necessary. In addition, the sample size used here is low. Although clinicopathological variables such as the IMDC risk group or pathologic type were corrected with multivariate analyses. Age and sex variation might have a larger impact on a smaller patient group size. Moreover, further studies are needed to determine whether IL-6 acts as a consistent biomarker in other immunotherapies, such as nivolumab monotherapy, nivolumab plus ipilimumab combination therapy, and combination therapies with vascular endothelial growth factor receptor inhibitor (pembrolizumab/lenvatinib and nivolumab/cabozantinib).

## 5. Conclusions

In conclusion, a high serum IL-6 level at baseline was related to poor clinical outcomes following first line Pembro/Axi treatment. The serum IL-6 level might be used as a circulating predictive biomarker for immunotherapy in patients with advanced RCC, including the selection of another treatment option rather than Pembro/Axi. Further studies are needed to determine whether it is a potential therapeutic target that should be utilized to enhance the efficacy of immunotherapy in RCC. The development of such a noninvasive blood-based predictive biomarker is important for the selection of optimal first line therapy for advanced RCC.

## Figures and Tables

**Figure 1 cancers-14-05985-f001:**
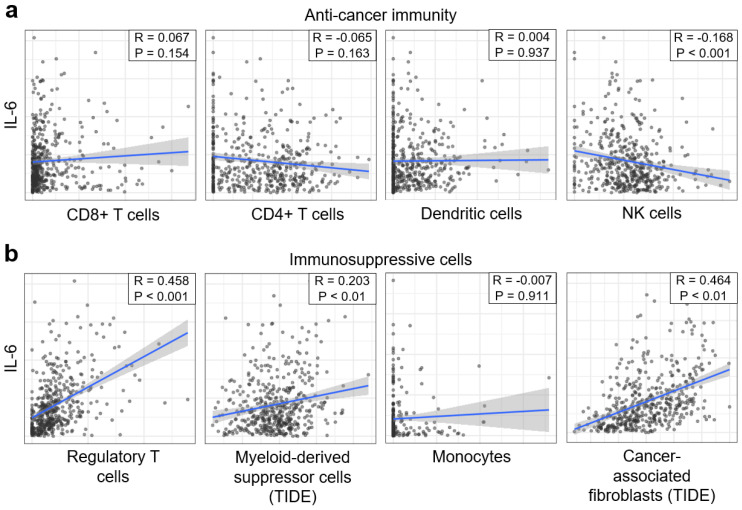
IL-6 levels are associated with tumor immunity in clear cell renal cell carcinoma. The correlations between IL-6 levels and immune cells or stromal cells were analyzed using clear cell RCC datasets (*n* = 533) from The Cancer Genome Atlas. (**a**) Correlation plots of IL-6 levels with anticancer immune cells: CD8+ T cells, CD4+ T cells, dendritic cells, and natural killer cells. (**b**) Correlation plots of IL-6 levels with regulatory T cells, myeloid-derived suppressor cells, monocytes, and cancer-associated fibroblasts were generated using Pearson’s correlation. The blue solid lines represent the regression curve, and the shaded areas represent the 95% confidence interval bands of the best fit. TIDE, tumor immune dysfunction, and exclusion.

**Figure 2 cancers-14-05985-f002:**
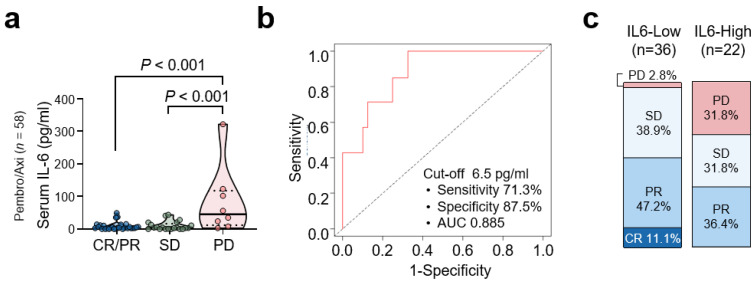
Baseline serum IL-6 levels and survival outcome with pembrolizumab/axitinib therapy. (**a**) Comparison of IL-6 levels according to the best response. Values were compared using ANOVA with Tukey’s post hoc test with respective *p*-values given in each plot. (**b**) Time-dependent receiver operator characteristic curve of serum IL-6 levels. (**c**) Bar charts showing the best response to therapy by IL-6 level.

**Figure 3 cancers-14-05985-f003:**
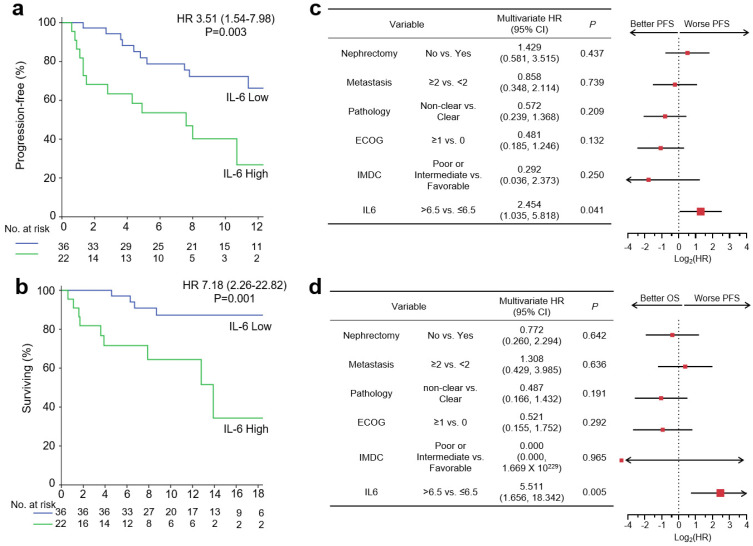
Baseline serum IL-6 levels and survival outcome with pembrolizumab/axitinib therapy. IL-6 > 6.5 pg/mL was defined as IL-6 High. (**a**) Kaplan–Meier curves showing PFS according to IL-6 level. (**b**) Kaplan–Meier curves showing OS according to IL-6 level. (**c**) Forest plots showing PFS according to nephrectomy, metastasis, pathology, Eastern Cooperative Oncology Group (ECOG) performance status, International Metastatic Renal Cell Carcinoma Data Consortium (IMDC) risk criteria, and IL-6 status. (**d**) Forest plots showing OS according to nephrectomy, metastasis, pathology, ECOG performance status, IMDC risk criteria, and IL-6 status. Hazard ratios and 95% confidence intervals are shown in survival curves and forest plots. Multivariate analyses were conducted using Cox proportional hazard regression. PFS, progression-free survival; OS, overall survival.

**Figure 4 cancers-14-05985-f004:**
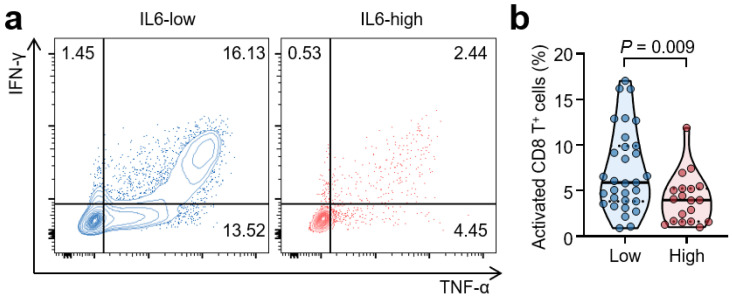
High serum IL-6 levels are associated with reduced CD8+ T cell function. (**a**,**b**) Representative flow cytometric plots and comparison of intracellular interferon-γ (IFN-γ) and tumor necrosis factor-α (TNF-α) levels from CD8+ T cells according to IL-6 status. T cells were stimulated with an anti-CD3 antibody for 24 h. Values were compared using a *t*-test, with respective *p*-values given in the plot.

**Table 1 cancers-14-05985-t001:** Demographics and disease characteristics at baseline.

Characteristic	Number of Patients, *n* (% of Total)
Age	
≥65	23 (39.7)
Male	42 (72.4)
ECOG performance status	
0	22 (37.9)
1	29 (50.0)
2	7 (12.1)
IMDC prognostic risk	
Favorable	11 (19.0)
Intermediate	29 (50.0)
Poor	18 (31.0)
Pathology	
Clear cell	41 (70.7)
Non-clear cell	17 (29.3)
No. of organs with metastases	
1	20 (34.5)
≥2	38 (65.5)
Sites of metastasis	
Lung	47 (81.0)
Lymph node	17 (29.3)
Bone	21 (36.2)
Liver	6 (10.3)
Previous nephrectomy	30 (51.7)

**Table 2 cancers-14-05985-t002:** Association between baseline IL-6 levels and patient characteristics.

Characteristic	IL-6 Low (*n* = 36)	IL-6 High (*n* = 22)	*p*-Value
Age			
Median (range), y	61 (39–82)	59 (47–83)	0.572
≥65, *n* (%)	13 (36.1)	10 (45.5)	0.480
Male, *n* (%)	28 (77.8)	14 (63.6)	0.242
ECOG performance status			0.316
0	16 (44.4)	6 (27.3)	
1	17 (47.2)	12 (54.5)	
2	3 (8.3)	4 (18.2)	
IMDC prognostic risk, *n* (%)			0.009
Favorable	9 (25.0)	2 (9.1)	
Intermediate	21 (58.3)	8 (36.4)	
Poor	6 (16.7)	12 (54.5)	
Pathology, *n* (%)			0.356
Clear cell	27 (75.0)	14 (63.6)	
Non-clear cell	9 (25.0)	8 (36.4)	
No. of organs with metastases, *n* (%)			0.366
0, 1	14 (38.9)	6 (27.3)	
≥2	22 (61.1)	16 (72.7)	
Sites of metastasis, *n* (%)			
Lung	28 (77.8)	19 (86.4)	0.418
Lymph node	7 (19.4)	10 (45.5)	0.035
Bone	9 (25.0)	12 (54.5)	0.023
Liver	4 (11.1)	2 (9.1)	0.806
Previous nephrectomy, *n* (%)	28 (53.8)	2 (33.3)	0.018

## Data Availability

The deidentified individual-level participant data can be provided to researchers upon request by emailing the corresponding authors. A detailed proposal for how the data will be used is required and will be reviewed on a case-by-case basis.

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
