# Peer review of "High Serum Levels of IL-6 Predict Poor Responses in Patients Treated with Pembrolizumab plus Axitinib for Advanced Renal Cell Carcinoma"

_cancers, 2022, doi:10.3390/cancers14235985_

Round 1

Reviewer 1 Report

The manuscript is written well and documented. I have some questions regarding this manuscript that must be addressed.  

1.     In this manuscript, the author did not show how IL6 makes the resistance RCC against pembrolizumab plus masitinib.

2.    There are many reports which show that High IL6 serum level induces poor prognosis, this may be true for most drugs including pembrolizumab and masitinib.

3.    The author is unable to show the driving force which regulates RCC.

4.    The author should identify the status of VHL expression/mutation in these patients.

Overall, this study is not innovative. A mountain of evidence suggests that IL6 induces drug resistance in renal clear cell carcinoma. 

Reviewer 2 Report

This is a comprehensive study on the prognostic role of IL-6 in Renal Carcinoma under FDA approved combination therapy of pembrolizumab (an immunotherapy) and axitinib (a tyrosine kinase inhibitor), for first-line treatment.

The study is well designed, and the manuscript is well written.

There are few concerns to address:

1.    There are several studies on prognostic value of IL-6 in Renal Carcinoma patients. It will be important to cite some.

2.    The TCGA dataset analyses showing positive correlation of IL-6 levels with regulatory T cells, myeloid-derived suppressor cells, and cancer- associated fibroblasts is exciting and strongly suggests the claim that elevated IL-6 levels may be associated with increased intra-tumoral immunosuppression in patients with kidney cancer.

It will be important to cite more recent advances on the IL-6 induced immunosuppression data associated with TME.

3.    Is any time of the day or metabolic status considered while taking the Serum samples?

4.    Did any of the patients listed undergo treatment withdrawal in the defined time frame of study?

5.    Despite the consideration to clinicopathological variables, such as IMDC risk group or pathologic type, by multivariate analyses, the sample size is low. The age and sex variation might have larger impact on a small patient size. The authors may mention it in the limitation section.

6.    To analyze the role of IL-6 signaling, could the authors also measure IL-6 receptor (and gp-130) levels in the same TCGA dataset, serum samples, or PBMCs?

7.    Line 114: The authors may specify the use of brefeldinA and monensin to end the stimulation for cytokine secretion.

8.    Line 76: The authors may rephrase the sentence.

Addressing these minor concerns would be satisfactory for publication of the manuscript as per my evaluation. 

Reviewer 3 Report

In this manuscript, the authors have described the possible role of serum IL-6 level in response to Pembro/Axi in RCC patients. The authors collected the blood samples from 58 RCC patients before treatment and measured the IL-6 level. Their results indicated that patients with high IL-6 levels had more IMDC poor-risk diseases, more frequent lymph nodes, and bone metastases. In addition, patients with high IL-6 level showed worse OS than those with low IL-6 level. The authors suggest that High IL-6 level was associated with reduced T-cell activity, resulting in lack of response to Pembro/Axi. Overall, the authors indicated that IL-6 level could served as a predictive biomarker for RCC patients selection of Pembro/Axi treatment. However, there are some questions still need to be answered:

1) Is there any difference of IL-6 level in male and female patients? Does sex affect the IL-6 baseline level?

2) Some studies suggested that IL-6 level serum could be altered by other factors, including extreme and long-term exercise (https://doi.org/10.2147/JIR.S281113). Do authors thought these factors could affect the conclusion in this article?

Reviewer 4 Report

Sangand co-workers present an interesting study on Renal cell carcinoma (RCC) to identify whether baseline serum IL-6 could serve as a predictive biomarker for Pembro/Axi treatment in RCC. 58 patients with advanced RCC were enrolled administered first-line Pembro/Axi and base line blood samples were analyzed using flow cytometry. The mean baseline serum IL-6 concentra was measured.

The autors conclude that overall, high baseline serum IL-6 levels are associated with worse survival outcome and reduced T-cell responses in Pembro/Axi-treated advanced RCC patients.

Generally the technical part of this work seems to be well conducted and performed. The procedures and techniques used are standard and appear appropriate. 

I believe that the results are of clinical relevance.

Author Response

Thank you very much for your comments. 

Round 2

Reviewer 1 Report

The manuscript is well-written and documented.

I am satisfied with the answer in 2nd round of review.

I will recommend publishing this article.